# Effectiveness of Hyperthermic Intraperitoneal Chemotherapy Associated with Cytoreductive Surgery in the Treatment of Advanced Ovarian Cancer: Systematic Review and Meta-Analysis

**DOI:** 10.3390/jpm13020258

**Published:** 2023-01-30

**Authors:** Maria Llueca, Maria Victoria Ibañez, Maria Teresa Climent, Anna Serra, Antoni Llueca

**Affiliations:** 1Department of Medicine, University CEU-Cardenal Herrera, 12006 Castellon, Spain; 2Department of Mathematics, IMAC, University Jaume I (UJI), 12006 Castellon, Spain; 3Multidisciplinary Unit of Abdominopelvic Oncology Surgery (MUAPOS), University General Hospital of Castellon, 12006 Castellon, Spain; 4Oncological Surgery Research Group (OSRG), Department of Medicine, University Jaume I (UJI), 12006 Castellon, Spain

**Keywords:** hyperthermic intraperitoneal chemotherapy (HIPEC), epithelial ovarian cancer, advanced stages, cytoreductive surgery, overall survival, disease-free survival, systematic revision, meta-analysis

## Abstract

Objective: The use of hyperthermic intraperitoneal chemotherapy (HIPEC) as a treatment for epithelial ovarian cancer remains controversial. Our study aims to analyze the overall survival and disease-free survival for the use of HIPEC as a treatment for advanced epithelial ovarian cancer after neoadjuvant chemotherapy. Methods: A systematic review and meta-analysis was carried out using *PubMed, Cochrane, Web of Science,* and *ClinicalTrials.gov*. A total of six studies were used, comprising a total of 674 patients. Results: The results of our meta-analysis of all studies analyzed together (observational and randomized controlled trials (RCT)) did not achieve significant results. Contrary to the OS (HR = 0.56, 95% IC = 0.33–0.95, *p* = 0.03) and DFS (HR = 0.61, 95% IC = 0.43–0.86, *p* < 0.01) of the RCT analyzed separately, a clear impact on survival was suggested. The subgroup analysis showed that studies making use of higher temperatures (≥42 °C) for a shorter period of time (≤60 min) achieved better results for both OS and DFS, as well as the use of cisplatin as the form of chemotherapy in HIPEC. Moreover, the use of HIPEC did not increase high-grade complications. Conclusions: The addition of HIPEC to cytoreductive surgery demonstrates an improvement in OS and DFS for patients with epithelial ovarian cancer in advanced stages, without an increase in the number of complications. The use of cisplatin as chemotherapy in HIPEC obtained better results.

## 1. Introduction

Ovarian carcinoma is considered a rare gynecological tumor and is the leading cause of death due to gynecological tumors in the female population. A large number of patients are diagnosed in advanced stages of the disease (International Federation of Gynecology and Obstetrics (FIGO) stage IIIC–IV) [1].

Peritoneal spread of ovarian cancer produces peritoneal metastasis, which is characteristic of advanced disease, and patients can remain in this stage for a long time [1]. The peritoneal cavity is not very accessible to systemic chemotherapy due to the peritoneal plasma barrier that makes it difficult for the drug to pass to the abdominal cavity [2,3,4,5].

Some clinical trials have described better survival in patients with ovarian cancer when chemotherapy is administered directly into the peritoneal cavity, although this procedure is associated with a high number of complications [6,7,8].

In addition, chemotherapy agents could be administered in association with hyperthermia (hyperthermic intraperitoneal chemotherapy (HIPEC)) to increase the effect of the drug with heat [9].

Treatment with cytoreductive surgery and HIPEC has shown promise in some types of tumors with peritoneal dissemination, and its application in ovarian cancer has aroused interest [10,11].

Many researchers have analyzed the effect of HIPEC when added to cytoreductive surgery in patients with primary or recurrent ovarian cancer. However, the technique has also been controversial because its evidence has been based mainly on retrospective observational studies, frequently with only one arm [12,13,14,15].

Van Driel et al. published in 2018 the first randomized controlled trial that investigated the effect of HIPEC on primary EOC after neoadjuvant treatment, in which 245 patients were randomized to receive HIPEC or not after cytoreductive surgery; disease-free survival (DFS) and overall survival (OS) were significantly higher in patients receiving HIPEC [16].

These results were recently reproduced in two randomized controlled trials. Cascales et al. in 2021 and Lim et al. in 2022 confirmed the results previously obtained by Van Driel, showing an improvement in both disease-free survival and overall survival in patients who had received HIPEC after cytoreductive surgery [17,18].

However, a previous comparative study also demonstrated that the addition of HIPEC to CRS in patients with platinum-sensitive recurrent ovarian cancer does not improve survival [19]. Hence, the subgroup (i.e., primary or recurrent) that can benefit most from HIPEC remains controversial.

Despite these data, controversies remain regarding the survival benefit and a possible increase in morbidity that the addition of HIPEC to cytoreductive surgery may produce [20].

In addition, the HIPEC (drug, timing, duration, temperature) regimens differed among investigations [20,21].

Therefore, we sought to provide comprehensive evidence for assessing the effects of HIPEC in upfront ovarian cancer treatment.

Based on the aforementioned factors, we aimed to quantitatively explore (meta-analysis) whether the addition of HIPEC in patients with primary ovarian cancer after neoadjuvant treatment could improve survival outcomes (OS or DFS). In addition, we systematically reviewed adverse events, morbidity, and mortality, as well as subgroup analyses derived from research to draw possible conclusions in terms of HIPEC administration regimens.

## 2. Methods

### 2.1. Search Strategy

A systematic review of the literature was conducted according to the PRISMA (Preferred Reporting Item for Systematic Reviews and Meta-analyses) systematic review guide (http://www.prismastatement.org, accessed on 1 June 2022) [22] to compare the results of HIPEC to cytoreductive surgery in ovarian cancer in advanced stages. This study was registered in an international database of prospectively registered systematic reviews in health, PROSPERO (CRD42022365031).

The series search was conducted in PubMed, Cochrane, Web of Science, and ClinicalTrials.gov between January 2011 and March 2022. Ovarian cancer and hyperthermic intraperitoneal chemotherapy were used as keywords.

The search terms used for all databases were as follows:

(“Ovarian neoplasm” OR “Ovarian cancer”) AND (“Hyperthermic intraperitoneal chemotherapy” OR “HIPEC”).

Clinical trials and observational studies that had a comparative arm were included.

### 2.2. Inclusion and Exclusion Criteria

According to the PICOS criteria (Population, Intervention, Comparison, Outcomes, and Study design) [22], studies were selected in our present meta-analysis according to the eligibility criteria reflected in Table 1, and the inclusion and exclusion criteria of the publications were established and are reflected in Table 2.

### 2.3. Selection Process

The selection of the articles was carried out by two researchers independently (M.LL. and M.T.C), and discrepancies between them were resolved by a third researcher (A.LL.).

The search was performed using the filters available in each of the databases: (1) Language: English and Spanish, (2) Date: Last 10 years (2011–2022), and (3) Type of study: Clinical trial.

### 2.4. Statistical Analysis

As a measure of association for OS and DFS between studies, the hazard ratio (HR) was used, and complications were studied using the relative risk (RR). Additional studies were carried out with subgroups, and the results are presented in different forest plots. Differences between patients treated with HIPEC and those treated with standard therapy were studied using random-effects analysis. Heterogeneity was assessed using I^2^ considering low heterogeneity < 25%, intermediate 50%, and high 75%. The results were considered statistically significant when *p* < 0.05 was obtained. All statistical analyses were carried out using RevMan 5 software (Review Manager (RevMan). Version 5.4. The Cochrane Collaboration, 2020)

## 3. Results

### 3.1. Selected Studies

Using the search strategy, a total of 1525 results were found. After filtering the articles with electronic tools and eliminating duplicates, selection was made by reading the title and abstract, and nine articles were extracted. Subsequently, an in-depth reading was performed, after which three articles were excluded. Six articles were ultimately included in the review. Figure 1 shows the selection process carried out due to the exclusion of the articles.

Among the included trials, three were randomized clinical trials (RCTs) [16,17,18] and three were observational studies with a control arm [23,24,25]. All six studies were published in the last five years [16,17,18,23,24], and two had a population > 100 patients [16,24]. A total of 674 patients were included in the present review. Detailed information on the included studies is summarized in Table 3 and Table 4.

Four of the selected studies showed the histological types included, which are summarized in Appendix A. Bias analyses are detailed in Appendix A. Observational studies show a greater bias as they are observational in nature but reach an acceptable level of evidence as they have a control arm. This is contrary to what happens in an RCT where the levels of evidence are much higher even when the patients have not been assigned blindly.

### 3.2. Overall Meta-Analyses of OS and DFS

Among the six studies included in our review, five reported OS data. The results obtained from each study are presented in Appendix A

Our pooled analysis indicated that patients who received HIPEC exhibited a significant improvement in OS compared to those treated with standard treatment (HR = 0.62, 95% IC = 0.38–0.99, *p* = 0.05) (Figure 2A).

On the other hand, a subgroup analysis was performed where the RCTs and the observational studies were compared separately. The RCTs showed an improvement in the OS of the group treated with HIPEC compared to the control group (HR = 0.56, 95% CI = 0.33–0.95, *p* = 0.03) (Figure 2B). Observational studies revealed no statistically significant differences in OS.

In addition, all studies included in the review provided DFS results. The global analysis of all the studies did not indicate statistically significant differences between the groups (Figure 3A). Subgroup analysis showed a significant improvement in DFS in RCTs of patients treated with HIPEC compared with patients treated with standard therapy (HR = 0.61, 95% CI = 0.43–0.86, *p* < 0.01) (Figure 3B). Observational studies did not show significant differences.

### 3.3. Subgroup Analyses

OS and DFS results from studies were collected and analyzed by grouping according to study type, publication year, number of participants, HIPEC regimen, HIPEC temperature, and HIPEC duration.

Studies using cisplatin as chemotherapy in HIPEC suggested an improvement in both OS (HR = 0.56, 95% CI = 0.33–0.95, *p* = 0.05) and DFS (HR = 0.61, 95% CI = 0.43–0.86, *p* < 0.01). Studies using carboplatin/cisplatin + paclitaxel showed a statistically significant increase in DFS in the control group compared to the HIPEC-treated group (HR = 1.95, 95% CI = 1.26–3.01, *p* < 0.01). One of the studies used paclitaxel, and that study reported an increase in DFS in the HIPEC-treated group (HR = 0.36, 95% CI = 0.21–0.68, *p* < 0.01).

Studies using HIPEC temperature < 42 °C and infusion time > 60 min indicated a significant increase in OS in the HIPEC-treated group but not in DFS. On the other hand, studies using temperature ≥ 42 °C and administration time ≤ 60 min revealed a significant increase in both OS and DFS of HIPEC-treated patients compared to controls. These results are detailed in Table 5.

### 3.4. Systematic Review of Adverse Events, Morbidity, and Mortality

Among the selected studies, four specified grade 3–5 complications in the first postoperative month according to the classification of the National Cancer Institute (NCI Common Terminology Criteria for Adverse Events, version 4) [17]. One of the studies could not be included in the analysis because it did not show the results of the group that used neoadjuvant treatment separately. The results are shown in Appendix A.

The analysis of the results did not show a statistically significant difference between the group treated with HIPEC and the control group (RR = 1.01, 95% CI = 0.71–1.45, *p* = 0.96). The results are reflected in Figure 4.

## 4. Discussion

Based on the general results of our meta-analysis, in patients with advanced ovarian cancer, cytoreduction surgery plus HIPEC improves both disease-free interval and overall survival compared to traditional cytoreduction surgery. When analyzing the subgroups of this study, we observed that this improvement in overall survival and disease-free time acquired significant values only when we considered the RCTs.

Although intraperitoneal chemotherapy has been shown to increase survival in advanced ovarian cancer [6], its use has not become widespread owing to technical problems with its administration through the catheter and its excessive toxicity on some occasions. HIPEC differs from postoperative intraperitoneal chemotherapy in that it is administered intraoperatively during debulking surgery, avoiding the side effects of postoperative intraperitoneal chemotherapy [26].

The administration of hyperthermic intraperitoneal chemotherapy during surgery exerts a direct cytotoxic effect due to the high temperature, which causes the denaturation of cellular proteins and simultaneously induces vasodilation, facilitating the entry of cytotoxic agents into the ovarian tumor [27,28].

In the 1980s, Sugarbaker introduced HIPEC for the treatment of digestive cancers, reporting increased survival and decreased relapses in patients with colorectal cancer [29].

It would not be until 2015 when Spilotis et al. published the first RCT designed for recurrences of ovarian cancer [30].

All studies included in this review were initiated with neoadjuvant chemotherapy and subsequently underwent interval cytoreduction surgery plus HIPEC. No studies were included where HIPEC was applied in primary cytoreduction surgery because

Very few studies have discussed this treatment modality. Only the first part of Lim’s study [17] discusses primary cytoreduction surgery plus HIPEC, and its results are discouraging, even though the series is small. We hope that the OVHIPEC 2 [31] study currently underway will shed light at the end of the recruitment.

The results of three RCTs evaluating neoadjuvant therapy plus cytoreduction surgery with HIPEC in patients with advanced ovarian cancer have recently been published (Van Driel et al. [16], Cheol Lim et al. [17], Cascales Campos et al. [18]).

The first RCT conducted by Van Driel et al. randomized 245 patients with stage III ovarian cancer and complete/optimal surgery who had previously received three cycles of neoadjuvant chemotherapy due to excessive tumor volume for primary surgery. Disease-free survival and overall survival for patients who had received HIPEC were superior to standard treatment (14.2 vs. 10.7 months and 45.7 vs. 33.9 months, respectively). This RCT showed scientific evidence that HIPEC improved both DFS and OS.

One of the criticisms of this work was that in the group that did not use HIPEC, OS was lower than in the control group of the GOG 172 study by Armstrong et al. [6]; this may have been due to the different inclusion criteria, as the latter only included patients who were candidates for primary surgery.

It should be noted that an important part of the variability in the results of advanced ovarian cancer surgeries could be because the FIGO classification is not useful to quantify the amount of abdominal tumor prior to surgery; therefore, it is difficult to compare results. The intraoperative quantification of the peritoneal cancer index is more in line with the reality of the amount of tumor in the abdomen and thus facilitates the comparison of results in different studies [32,33,34].

Some studies with previous negative or inconclusive results in the primary treatment of ovarian cancer have also been described. Mendivil et al. [24] reported an increase in disease-free survival but did not provide an increase in overall survival. This can be explained because the patients to whom HIPEC had been applied had a much shorter follow-up period than those in the control group. In addition, both groups received six cycles of adjuvant chemotherapy after surgery, and a minilaparotomy was performed to infuse HIPEC within the first three weeks after cytoreduction surgery, which differs from the neoadjuvant approach, understood as the administration of 3–4 cycles prior to surgery. Therefore, their results should not be extrapolated and exceed the purpose of this meta-analysis.

The analysis of the subgroups of this study based on the type of study, number of patients, year of publication of the study, and HIPEC administration regimen (duration, drug, and temperature) also support the results of the meta-analysis, although only the RCTs prove them, statistically speaking.

The drugs used in HIPEC have been the subject of debate and controversy in recent years. Various drugs have been used (cisplatin, carboplatin, paclitaxel, doxorubicin, oxaliplatin, mitomycin, etc.) with a single pharmacokinetic property in common, that of remaining in the abdominal cavity, exerting its effect locally without being absorbed systemically [35].

The results of this meta-analysis suggest a benefit of the use of cisplatin at doses ranging between 75 and 100 mg/m^2^ as a drug for HIPEC. Platinum compounds show good tissue penetration, especially when applied at high temperatures [36].

In addition, when analyzing the subgroups of the temperature reached by the HIPEC administration solution, we observed greater efficacy at high temperatures (≥42 °C) and an exposure time ≤ 60 min, showing an increase in both OS and DFS in comparison with a lower temperature and exposure time, which only indicated an improvement in the DFS.

These results do not agree with other systematic reviews [37] that showed the same effectiveness for lower temperatures and longer exposure times. However, in this study, most of the included studies were observational, including both primary surgeries and surgeries after recurrence. In addition, they had great heterogeneity in terms of drugs, temperatures and exposure times, but all the studies coincided in using the open technique for the administration of HIPEC, except in Mendivil’s study [24] where the HIPEC was administered through a closed technique (minilaparotomy).

Based on our results, the greatest efficacy of HIPEC in ovarian cancer would be obtained with high temperatures and with an exposure time of one hour.

Similar to other authors [37], the morbidity reported in this meta-analysis of the series that includes the rate of major complications in their results does not seem to influence the result, as there are no differences between the control and experimental arms. Therefore, the complications resulting from HIPEC may not be a risk factor for survival. In addition, the rate of complications in this study is within the percentiles of the “quality indicators” defined in cytoreduction surgery for advanced ovarian cancer without HIPEC. However, it is worth noting the high degree of complications reported by Lim in his work; these results are outside the quality standards defined for this surgery, but even so, there are no differences between the two arms of the study [32].

In the present meta-analysis, there are a number of limitations. First, HIPEC inclusion criteria and drug regimens are varied. However, these are reduced if we focus on RCTs in which the use of cisplatin predominates. Second, not all studies describe the peritoneal cancer index as a quantification method and as an independent prognostic factor for advanced stages of ovarian cancer. Third, the definition of optimal and suboptimal surgery differs depending on the description of the different authors. Some define optimal surgery as that with tumor residues < 1 cm, and others refer to a size < 2.5 cm. These differences may produce variations in the survival results, as it has already been shown that one of the greatest prognostic factors in the treatment of advanced ovarian cancer is the residual tumor [6].

On the other hand, this meta-analysis is the first to include the results of three RCTs which gives the results of this study the highest level of evidence to date on the application of HIPEC in the treatment of ovarian cancer in the first line.

## 5. Conclusions

The present meta-analysis demonstrates an increase in disease-free interval and overall survival in patients with advanced ovarian cancer undergoing interval surgery plus HIPEC after neoadjuvant therapy.

## Figures and Tables

**Figure 1 jpm-13-00258-f001:**
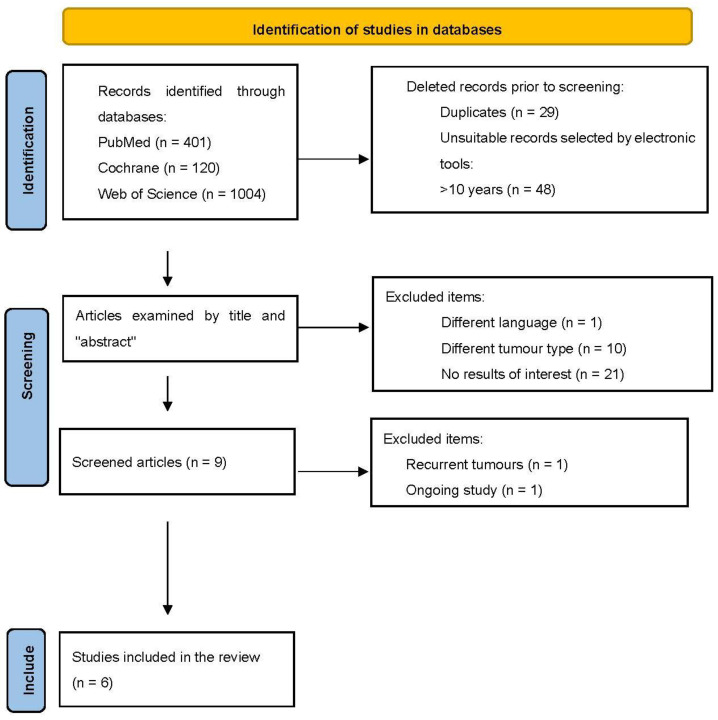
PRISMA diagram showing the selection process of the articles included in this study.

**Figure 2 jpm-13-00258-f002:**
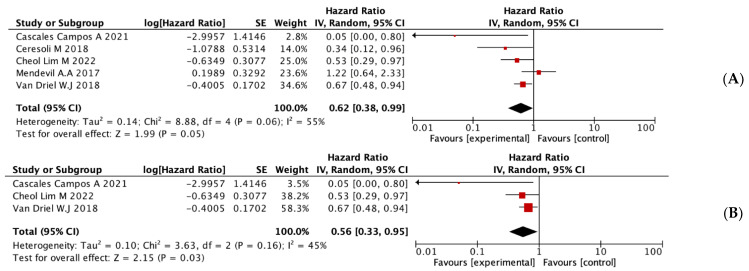
Statistical analysis of OS. (**A**) Results obtained from all studies included in the review. (**B**) Results obtained from the subgroup analysis of the RCTs. [16,17,18,23,24].

**Figure 3 jpm-13-00258-f003:**
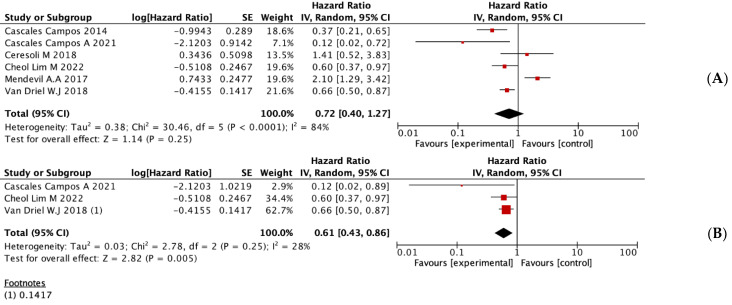
Statistical analysis of DFS. (**A**) Results obtained from all studies included in the review. (**B**) Results obtained from the subgroup analysis of the RCTs. [16,17,18,23,24,25].

**Figure 4 jpm-13-00258-f004:**
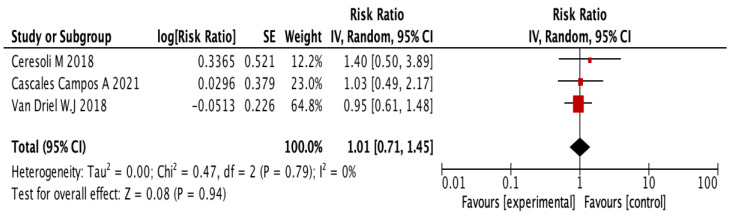
Analysis results of grade III–V complications in the first postoperative month. [16,23,25].

**Table 1 jpm-13-00258-t001:** PICOS criteria.

Population	Women with ovarian epithelial carcinoma in FIGO stages III–IVa.
Intervention	Neoadjuvant chemotherapy + Cytoreductive surgery + HIPEC ± Adjuvant chemotherapy
Comparison	Neoadjuvant chemotherapy + Cytoreductive surgery ± Adjuvant chemotherapy
Outcome	Overall survival (OS)
	Disease free survival (DFS)
	Complications
Study design	Clinical trials
	Observational studies

**Table 2 jpm-13-00258-t002:** Inclusion and exclusion criteria.

Inclusion Criteria	Exclusion Criteria
(1) Clinical trial and observational studies (2) Publication 2011–2022 (3) Spanish and English (4) Epithelial ovarian cancer (5) Advanced stages (FIGO III–IVa) (6) Upfront treatment (only neoadjuvant chemotherapy and IDS)	(1) Nonepithelial ovarian cancer (2) Extra-abdominal metastases (3) Absence of postoperative morbidity and mortality data (4) Absence of complete debulking surgery (5) No neoadjuvant chemotherapy (6) Studies without control group

**Table 3 jpm-13-00258-t003:** Main characteristics of the selected studies.

	Publication Date	Type of Study	Number of Patients	Control Arm	Experimental Arm	NACT Scheme	HIPEC		
							HIPEC Scheme	Temperature	Duration
Cheol Lim M. et al. [17]	2022	RCT	77	N = 43 NACT + CRS+ ASC	N = 34 NACT + CRS + HIPEC + ASC	Carboplatin (5 mg/mL/min) + Paclitaxel (175 mg/m^2^)	Cisplatin (75 mg/m^2^)	41.5 °C	90 min
Cascales Campos P. et al. [18]	2021	RCT	71	N = 36 NACT+ CRS+ ASC	N = 35 NACT + CRS + HIPEC + ASC	Carboplatin (AUC 5) + Paclitaxel (175 mg/m^2^)	Cisplatin (75 mg/m^2^)	42–43 °C	60 min
Van Driel W.J. et al. [16]	2018	RCT	245	N = 123 NACT + CRS + ASC	N = 122 NACT + CRS + HIPEC + ASC	Carboplatin (5 mg/m^2^) + Paclitaxel (175 mg/m^2^)	Cisplatin (100 mg/m^2^)	40 °C	90 min
Mendevil A. et al. [24]	2017	CC	138	N = 69 NACT + CRS + ASC	N = 69 NACT + CRS + HIPEC	NE	Carboplatin (6 AUC) + Paclitaxel (80 mg/m^2^)	41.5 °C	90 min
Ceresoli M. et al. [23]	2018	CC	56	N = 28 NACT + CRS + ASC	N = 28 NACT + CRS + HIPEC + ASC	Carboplatin + Paclitaxel	Cisplatin (100 mg/m^2^) + Paclitaxel (175 mg/m^2^)	41.5 °C	90 min
Cascales Campos P. et al. [25]	2014	CC	87	N = 35 NACT + CRS + ASC	N = 52 NACT + CRS + HIPEC + ASC	NE	Paclitaxel 60 mg/m^2^	42 °C	60 min

RCT, Randomized clinical trial; CC, Cases and controls; n, Population; NACT, Neoadjuvant chemotherapy; CRS, Cytoreductive surgery; HIPEC, Hyperthermic intraperitoneal chemotherapy; ASC, Adjuvant systemic chemotherapy; NE, not specified.

**Table 4 jpm-13-00258-t004:** Characteristics of the patients included in the studies.

	Arm	Patients	Age Median (Range)	FIGO n (%)	PCI Median (Range)	Median Intraoperative Time Min (Range)	Cytoreductive Surgery n (%)
Cheol Lim M. et al. [17]	Control	n = 43	54 (48–61)	III n = 17 (36.5) IV n = 26 (60.5)	>6 (6–10)	384 (328–437)	CC-0 37 (86) CC-1 6 (14)
	Experimental	n = 34	55 (47–64)	III n = 15 (44.1) IV n = 19 (55.9)	>6 (6–10)	506.5 (449–570)	CC-0 27 (79.4) CC-1 7 (20.6)
Cascales Campos P. et al. [18]	Control	n = 36	65.5 (40–75)	III n = 30 (83.5) IV n = 6 (16.7)	7 (2–29)	220 (140–345)	CC-0 32 (88.9) CC-1 4 (11.1)
	Experimental	n = 35	56 (29–75)	III n = 33 (94.3) IV n = 2 (5.7)	10 (2–22)	300 (220–490)	CC-0 33 (94.3) CC-1 2 (5.7)
Van Driel W.J. et al. [16]	Control	n = 123	63 (56–66)	III n = 123	NE	192 (153–251)	CC-0 82 (67) CC-1 24 (20)
	Experimental	n = 122	61 (55–66)	III n = 122	NE	338 (299–426)	CC-0 84 (69) CC-1 22 (18)
Ceresoli M. et al. [23]	Control	n = 28	61.55	III n = 20 (71.4) IV n = 8 (28.6)	6,36	194	CC-0 n = 23 (92.9) CC-1 n = 1 (36.6)
	Experimental	n = 28	58.99	III n = 22 (78.6) IV n = 6 (21.4)	8,25	533	CC-0 n = 23 (92.9) CC-1 n = 1 (3.6)
Mendevil A.A et al. [24]	Control	n = 69	62.9	III n = 61 (88.4) IV n = 8 (11.6)	NE	NE	CC-0 n = 64 (92.7) CC-1 n = 5 (7.3)
	Experimental	n = 69	59.8	III n = 62 (89.9) IV n = 7 (10.1)	NE	NE	CC-0 n = 69 (100)
Cascales Campos P. et al. [25]	Control	n = 35	57 (29–73)	III n = 29 (83) IV n = 6 (17)	6 (3–16)	148,8	CC-0 n = 35 (100)
	Experimental	n = 52	57 (34–79)	III n = 47 (90) IV n = 5 (10)	9 (3–26)	360,8	CC-0 n = 52 (100)

CC-0, complete cytoreduction (absence of tumor at the macroscopic level); CC-1, residual tumor ≤ 2.5 mm; n, number of patients; NE, not specified; PCI, peritoneal cancer index.

**Table 5 jpm-13-00258-t005:** Analyses of subgroups.

FEATURES	OVERALL SURVIVAL (OS)				DISEASE FREE SURVIVAL (DFS)			
	Number of Studies	HR (95% IC)	*p* Value	Heterogeneity	Number of Studies	HR (95% IC)	*p* Value	Heterogeneity
TYPE OF STUDY								
RCTS	3	0.56 (0.33–0.95)	0.03	45%	3	0.61 (0.43, 0.86)	<0.01	28%
OBSERVATIONAL	3	0.69 (0.20–2.39)	0.56	79%	3	1.02 (0.30–3.43)	0.97	91%
YEAR OF PUBLICATION								
≤2018	3	0.71 (0.41–1.25)	0.24	58%	4	0.90 (0.43–1.89)	0.77	88%
>2018	2	0.24 (0.03–2.15)	0.2	75%	2	0.34 (0.08–1.54)	0.16	78%
NUMBER OF PATIENTS								
<100	4	0.44 (0.26–0.77)	<0.01	33%	4	0.52 (0.27–1.00)	0.05	63%
>100	2	0.85 (0.45–1.50)	0.57	62%	2	1.16 (0.37–3.60)	0.80	88%
HIPEC REGIME								
CISPLATIN	3	0.56 (0.33–0.95)	0.03	45%	3	0.61 (0.43–0.86)	<0.01	28%
CARBOPLATIN/CISPLATIN + PACLITAXEL	2	0.69 (0.2–2.39)	0.56	76%	2	1.95 (1.26–3.01)	<0.01	0%
PACLITAXEL	1	NR			1	0.36 (0.21–0.68)	<0.01	NA
HIPEC TEMPERATURE								
<42	4	0.68 (0.52–0.88)	<0.01	46%	4	1.00 (0.53–1.86)	1.00	85%
≥42	2	0.05 (0.00–0.8)	0.03	NA	2	0.32 (0.16–0.66)	<0.01	11%
HIPEC DURATION								
>60 MIN	4	0.68 (0.52–0.88)	<0.01	46%	4	1.00 (0.53–1.86)	1.00	85%
≤60 MIN	2	0.05 (0.00–0.8)	0.03	NA	2	0.32 (0.16–0.66)	<0.01	11%

RCT, Randomized clinical trial; HIPEC, Hyperthermic intraperitoneal chemotherapy; HR, Hazard ratio; CI, Confidence interval.

## Data Availability

The data that support the findings of this study are available from the corresponding author, [A.LL.], upon reasonable request.

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
