# Peer review of "Effectiveness of Hyperthermic Intraperitoneal Chemotherapy Associated with Cytoreductive Surgery in the Treatment of Advanced Ovarian Cancer: Systematic Review and Meta-Analysis"

_jpm, 2023, doi:10.3390/jpm13020258_

Round 1
Reviewer 1 Report
This is an interesting and well written metanalysis. I have only a few suggestions to improve the strength of the paper.
Abstract, Results: the first sentence in unclear. The authors atate that results of their meta-analysis of all studies (observational and randomized controlled trials (RCT)) did not achieve significant results. However, a significant P value is reported for both overall (P=0,03) and disease-free survival (P <0,01). Please, clarify.
Peritoneal metastases is the term currently most used, as it suggests a curable disease, instead of peritoneal carcinomatosis.
Introduction: more studies on intraperitoneal (normothermic chemotherapy in ovarian cancer may be referenced, in addition to ref. N. 6 (Armstrong et al.)
Methods, Selection process: please, specify which researchers carried out the selection of the articles, and which researcher resolved the discrepancies between them.
Methods, Statistics: did all the selected studies reported the hazard ratio for overall and disease-free survival or was it calculated by the authors ? and how ?
Results: please, edit reference number on page 4, line 156. It appears to be (21-23), not (1-23)
Table 5 (Supplementary material 1): head row is in Spanish language.
Results: in their study Cheol Lim M. et al. et al report grade 3-4 morbidity rates of 93.5% in experitmental group and 87% in control group. These figures are not in line with the other studies included in the present review/metanalysis and all the literature about CRS/HIPEC. This raises doubts about the opportunity to include the paper in the morbidity analysis (and the study itself).
References 30 and 31 are not needed
Author Response
Thank you very much for your work. I think that with your excellent review this work has improved a lot.
Reviewer 1
This is an interesting and well written metanalysis. I have only a few suggestions to improve the strength of the paper.
1.- Abstract, Results: the first sentence in unclear. The authors atate that results of their meta-analysis of all studies (observational and randomized controlled trials (RCT)) did not achieve significant results. However, a significant P value is reported for both overall (P=0,03) and disease-free survival (P <0,01). Please, clarify.
Answer:Thanks for your appreciation. Indeed, the reviewer is right, the sentence is not well understood. What it really means is that the joint analysis of the two types of study (observational and RCT) analyzed together do not reveal statistically significant differences, but not the analysis of the RCTs separately, which do show significantly significant data. clear, as reported by the reviewer.
We have corrected it by modifying the first sentence
2.-Peritoneal metastases is the term currently most used, as it suggests a curable disease, instead of peritoneal carcinomatosis.
Answer: Thank you for your suggestion. We have modified the term in the manuscript
Introduction: more studies on intraperitoneal (normothermic chemotherapy in ovarian cancer may be referenced, in addition to ref. N. 6 (Armstrong et al.)
Answer: We haved already added several references in this item. Thanks again
Methods, Selection process: please, specify which researchers carried out the selection of the articles, and which researcher resolved the discrepancies between them.
Answer: We have already stated the researchers as demanded
Methods, Statistics: did all the selected studies reported the hazard ratio for overall and disease-free survival or was it calculated by the authors ? and how ?
Answer: Some studies reported the hazard ratio and others were calculated, extracting the data from the study and with the support of RevMan 5 software (Review Manager (RevMan). Version 5.4. The Cochrane Collaboration, 2020), as already stated in methods section.
Results: please, edit reference number on page 4, line 156. It appears to be (21-23), not (1-23)
Answer: Thank you , already edited
Table 5 (Supplementary material 1): head row is in Spanish language.
Answer: Already corrected,thanks again.
Results: in their study Cheol Lim M. et al. et al report grade 3-4 morbidity rates of 93.5% in experitmental group and 87% in control group. These figures are not in line with the other studies included in the present review/metanalysis and all the literature about CRS/HIPEC. This raises doubts about the opportunity to include the paper in the morbidity analysis (and the study itself).
Answer: Totally agree with the reviewer, but the data from this study are published in a high-impact journal and although its results in terms of morbidity from this type of surgery and treatment are not good, we cannot fail to reflect them in this systematic review. Not without making it clear to the reader of the work that this percentage of complications, in this type of surgery, is not adequate and exceeds the quality standards (*)defined in world literature, as our reviewer very well indicates.
(*)Llueca A, Serra A, Climent MT, Segarra B, Maazouzi Y, Soriano M, Escrig J; on behalf MUAPOS Working Group. Outcome quality standards in advanced ovarian cancer surgery. World J Surg Oncol. 2020 Nov 25;18(1):309. doi: 10.1186/s12957-020-02064-7. Erratum in: World J Surg Oncol. 2020 Dec 7;18(1):323. PMID: 33239057; PMCID: PMC7690155.
References 30 and 31 are not needed
Answer: Thank you for your assessment, we have deleted the reference of the FIGO classification for ovarian cancer, however we believe that the reference of the first RCT assessing the efficacy of HIPEC in recurrences of ovarian cancer makes all the sense in this introduction. I would ask our reviewer to allow us to keep it so as not to weaken from the introduction.
Reviewer 2 Report
General comments:
The topic of this paper is both important and interesting.
The advanced stages of epithelial ovarian cancer are, unfortunately, the most frequent stages and the survival is poor. Therefore, many initiatives have been taken to improve survival but only by improving survival insignificant.
HIPEC has been used in different centers globally but the results have been rather confusing as we have been missing an overview because of difficulties in comparing the different research modalities. Therefore, a systematic review and meta-analysis is important.
It is noteworthy that the three RCTs demonstrate a significant impact on overall survival and disease-free survival. However, I think that some methodological uncertainties have to be clarified.
Specific comments:
Abstract:
The abstract is well-written and contains the important information. However, I think it has to be clarified that the six included studies only include HIPEC + neoadjuvant chemotherapy (NACT) and do not include HIPEC + up-front surgery.
Introduction:
The introduction is well-written and gives a fine and balanced introduction to the topic of the paper.
Methods:
The first part of the methods section gives a very thorough description of the literature search and the PRISMA guideline shows carefully the selection of papers for the review.
The inclusion and exclusion criteria are defined; however, inclusion criterion #6 is ‘upfront treatment’. In my opinion, upfront treatment is primary debulging surgery (PDS) / cytoreductive surgery (CRS) associated with adjuvant chemotherapy. According to table 3, the six included protocols are all HIPEC in combination with NACT, and none of the studies are upfront treatment combined with HIPEC. This has to be clarified.
Results:
As mentioned in the previous section, table 3 has to be commented on.
Moreover, in the results section a clearer distinction between published and supplementary material has to be made. However, the immediate confusion might be due to the PDF-conversion of your text file.
In table 5 (supplementary) the headings have to be translated into English.
Fig. 2 (supplementary) would gain by more explaining text.
In table 6 and Fig. 3 and 4 consider to omit Cascales Campos P. et al. 2014 as the contribution of this study to you results seems scanty.
Discussion:
In the discussion, you have to mention that the six included studies all include NACT + HIPEC and no studies with upfront treatment + HIPEC have been included in your review. You might not have found any relevant studies using this regime but then you have to discuss this.
No matter what, the upfront treatment with PDS / CRS + HIPEC must be preferable instead of patients estimated with a heavy tumor burden and therefore treated with NACT before CRS; discuss this.
Moreover, you have to mention / discuss the applied HIPEC technique. In the literature both an open and a closed HIPEC technique have been described buy you don’t mention which technique that has been used in the included studies in your review. This should be done.
In the study by Mendivil et al. you wrote: “…. and a minilaparotomy was performed to infuse HIPEC within the first 3 weeks after cytoreduction surgery, which differs from the neo adjuvant approach, understood as the administration of 3-4 cycles prior to surgery”. Do you think this study is comparable to the other included studies? If not, you have to exclude this study from your review.
Author Response
Thank you very much for your work. I think that with your excellent review this work has improved a lot.
Reviewer 2
Comments and Suggestions for Authors
General comments:
The topic of this paper is both important and interesting.
The advanced stages of epithelial ovarian cancer are, unfortunately, the most frequent stages and the survival is poor. Therefore, many initiatives have been taken to improve survival but only by improving survival insignificant.
HIPEC has been used in different centers globally but the results have been rather confusing as we have been missing an overview because of difficulties in comparing the different research modalities. Therefore, a systematic review and meta-analysis is important.
It is noteworthy that the three RCTs demonstrate a significant impact on overall survival and disease-free survival. However, I think that some methodological uncertainties have to be clarified.
Specific comments:
Abstract:
The abstract is well-written and contains the important information. However, I think it has to be clarified that the six included studies only include HIPEC + neoadjuvant chemotherapy (NACT) and do not include HIPEC + up-front surgery.
Answer:Thank you for your appreciation, we have made the amendments to clarify this.
Introduction:
The introduction is well-written and gives a fine and balanced introduction to the topic of the paper.
Methods:
The first part of the methods section gives a very thorough description of the literature search and the PRISMA guideline shows carefully the selection of papers for the review.
The inclusion and exclusion criteria are defined; however, inclusion criterion #6 is ‘upfront treatment’. In my opinion, upfront treatment is primary debulging surgery (PDS) / cytoreductive surgery (CRS) associated with adjuvant chemotherapy. According to table 3, the six included protocols are all HIPEC in combination with NACT, and none of the studies are upfront treatment combined with HIPEC. This has to be clarified.
Answer: Thank you very much for your observation.
In addition, exclusion criterion 5 already establishes that non-neoadjuvant therapy is a criterion that excludes the study.
In order to recruit a larger number of studies, the authors have considered up front treatment of advanced ovarian cancer both PDS+adjuvant Chemotherapy and Neoadjuvant+IDS as they are two forms of treatment for initial ovarian cancer with similar results, already described. widely in the literature. In any case, it has been clarified in the inclusion criteria table as indicated by the reviewer.
Results:
As mentioned in the previous section, table 3 has to be commented on.
Answer: Already answered before.
Moreover, in the results section a clearer distinction between published and supplementary material has to be made. However, the immediate confusion might be due to the PDF-conversion of your text file.
Answer: Yes you’re a re right. We have difficulties when converting word document to PDF. We will try to amend this
In table 5 (supplementary) the headings have to be translated into English.
Answer: Yes tour are right, sorry for the error. Already corrected
Fig. 2 (supplementary) would gain by more explaining text.
Answer: Thank you, already done
In table 6 and Fig. 3 and 4 consider to omit Cascales Campos P. et al. 2014 as the contribution of this study to you results seems scanty.
Answer: Dear reveiewer, I feel the obligation to include the work of Cascales from 2014 because although it seems scanty in terms of his contribution to this work, this author is one of the pioneers and leaders in HIPEC treatment of ovarian cancer after neoadjuvant therapy. The data from him is what gave rise to the subsequent RCT that he himself published in 2022.
Therefore, I would ask you to allow me to keep this work as part of the systematic review and its number of patients is not negligible (87 patients) that confer greater statistical power to the investigation
Discussion:
In the discussion, you have to mention that the six included studies all include NACT + HIPEC and no studies with upfront treatment + HIPEC have been included in your review. You might not have found any relevant studies using this regime but then you have to discuss this.
No matter what, the upfront treatment with PDS / CRS + HIPEC must be preferable instead of patients estimated with a heavy tumor burden and therefore treated with NACT before CRS; discuss this.
Answer: Thank you again. We have already introduced a paragraph to discuss this item in the discussion section.
Moreover, you have to mention / discuss the applied HIPEC technique. In the literature both an open and a closed HIPEC technique have been described buy you don’t mention which technique that has been used in the included studies in your review. This should be done.
Answer: All studies included in this systematic revison employed open hipec technique. This has already stated in the manuscript
In the study by Mendivil et al. you wrote: “…. and a minilaparotomy was performed to infuse HIPEC within the first 3 weeks after cytoreduction surgery, which differs from the neo adjuvant approach, understood as the administration of 3-4 cycles prior to surgery”. Do you think this study is comparable to the other included studies? If not, you have to exclude this study from your review.
Answer: Thanks for the observation. Mendivil's work, although in a different way, reproduces the HIPEC technique in ovarian cancer and also has favorable results. But it is important to note that in this work, HIPEC is performed 3 weeks after surgery, but the patient has already been treated with intravenous chemotherapy as neoadjuvant. It is important to highlight that although its technique is different, it also produces a benefit by delaying the time of appearance of the recurrence. For this reason, I think it has to be included in this study
Round 2
Reviewer 2 Report
I think your paper has improved assorting to my comments.
However, you wrote. "but all the studies coincided in using the open tech-nique for the administration of HIPEC". In the study by Mendivil et al. the HIPEC procedure was performed 3 weeks after surgery. I suppose, this was performed with a closed technique? Please correct.
Author Response
Thanks for your observation.
Your are right,Indeed, in the Mendivil study, the hipec was administered through a closed technique (minilaparotomy).
This was already corrected
Thanks again for your kind review process